# Aneuploidy and DNA Methylation as Mirrored Features of Early Human Embryo Development

**DOI:** 10.3390/genes11091084

**Published:** 2020-09-17

**Authors:** Ekaterina N. Tolmacheva, Stanislav A. Vasilyev, Igor N. Lebedev

**Affiliations:** Research Institute of Medical Genetics, Tomsk National Research Medical Center, 634050 Tomsk, Russia; stanislav.vasilyev@medgenetics.ru (S.A.V.); igor.lebedev@medgenetics.ru (I.N.L.)

**Keywords:** aneuploidy, DNA methylation, human embryogenesis, epigenetic reprogramming

## Abstract

Genome stability is an integral feature of all living organisms. Aneuploidy is the most common cause of fetal death in humans. The timing of bursts in increased aneuploidy frequency coincides with the waves of global epigenetic reprogramming in mammals. During gametogenesis and early embryogenesis, parental genomes undergo two waves of DNA methylation reprogramming. Failure of these processes can critically affect genome stability, including chromosome segregation during cell division. Abnormal methylation due to errors in the reprogramming process can potentially lead to aneuploidy. On the other hand, the presence of an entire additional chromosome, or chromosome loss, can affect the global genome methylation level. The associations of these two phenomena are well studied in the context of carcinogenesis, but here, we consider the relationship of DNA methylation and aneuploidy in early human and mammalian ontogenesis. In this review, we link these two phenomena and highlight the critical ontogenesis periods and genome regions that play a significant role in human reproduction and in the formation of pathological phenotypes in newborns with chromosomal aneuploidy.

## 1. Introduction

Genome stability is an integral feature of all living organisms. The need to maintain the normal karyotype is best illustrated by the fatal consequences of chromosomal disorders at the organism level. Aneuploidy is the most common cause of fetal death in humans. Among spontaneous abortions during the first trimester of pregnancy, aneuploid karyotypes are present in almost half of cases [1], and most of the observed aneuploidies are associated with errors in chromosome segregation occurring during oogenesis. In addition, the analysis of blastocysts, which became possible after the development of Preimplantation Genetic Testing for Aneuploidies (PGT-A) within in vitro fertilization (IVF) procedures, showed the presence of a large number of mosaic aneuploidies in preimplantation embryos [2,3,4]. The normal karyotype is most critical during the embryonic stage of development due to the importance and complexity of the differentiation, growth, and developmental processes that occur during this stage [5]. In other mammals, the frequency of aneuploidies has not been studied as well as in humans, but it has been shown that, in the non-human primates rhesus macaques, the frequency of aneuploid blastocysts is comparable to that in humans [6] and cows [7]. However, in mice, aneuploidies are not common; embryonic aneuploidy is observed in approximately 1% of blastocysts, depending on the line [8].

It is significant that the timing of rapid increases in the frequency of numerical chromosome abnormalities coincides with waves of global epigenetic reprogramming, mainly DNA methylation and demethylation processes, in mammals. In general, DNA methylation is stable in somatic tissues. However, the patterns and levels of DNA methylation dynamically change during development. During gametogenesis and early embryogenesis, the genome undergoes two waves of epigenetic reprogramming to allow for the development of the next generation. The two waves of DNA methylation reprogramming include both specific and common mechanisms, and failure of these mechanisms can critically affect genome stability as a whole, including the segregation of chromosomes during cell division.

DNA methylation has two roles: it regulates gene expression and chromatin structure [9]. Abnormal methylation due to errors in the reprogramming process can potentially lead to aneuploidy. On the other hand, the presence of an additional chromosome, or its loss, can affect the global genome methylation level. The associations of these two phenomena are better studied in the context of carcinogenesis [10,11], but here we consider the relationship of DNA methylation and aneuploidy in early human and mammalian ontogenesis, with minimal references to the field of carcinogenesis. In this review, we link these two phenomena and highlight the ontogenesis periods and genome regions that are crucial for human reproduction and for the formation of pathological phenotypes in newborns with chromosomal aneuploidy.

## 2. Incidence of Aneuploidy in Ontogenesis

Aneuploidy occurs at all stages of human ontogenesis, but its greatest incidence occurs during gametogenesis and early embryogenesis before implantation. In addition, in recent years, data on the high probability of mitotic errors during the initial stages of the division of oogonia have accumulated [12].

A high aneuploidy frequency is observed in mammalian oocytes, including human oocytes. Aneuploidy in oocytes depends on the age of the mother, with a higher incidence of aneuploidy at a very young age (less than 20 years old (yo)) and at an older age (more than 33 yo); age is the main factor that determines the shape of the human fertility curve [13]. Age-related aneuploidy is also common in other mammals: in the oocytes of old mice (aged 60–70 weeks), a six-fold increase in the frequency of hyperploidy was recorded [14], and the frequency of aneuploidy was three times higher in the oocytes of cows of older reproductive age with a reduced number of antral follicles than in those of normal cows [15].

In spermatozoa, aneuploidies are also found, but at a much lower frequency. In a group of healthy fertile men, disomy per autosome was approximately 0.1% [16]. Most often, in spermatozoa, disomy of chromosome 21 and sex chromosomes occurs [17]. In men with infertility, the aneuploidy frequency is significantly higher and can reach 18% [18]. Sex chromosome aneuploidies are most often of paternal origin [19].

A high aneuploidy frequency is also observed at the cleavage stage. Approximately half of human preimplantation embryos have aneuploidy, mostly of maternal origin. In addition, the cleavage stage is characterized by mitotic errors that lead to the appearance of mosaic karyotypes [20]. Early studies showed that the frequency of blastocysts with chromosomal mosaicism varied from 17.6% to 95% [3,21,22,23,24]. Moreover, these studies showed a direct correlation of mosaicism with maternal age [3]. According to the latest estimates, the level of mosaic karyotypes in human preimplantation embryos is significantly lower—from 4% to 22% [20]. The effect of maternal age on the frequency of mosaic embryos is also controversial, since it has not been confirmed in other studies [25,26]. In the non-human primates rhesus macaques, the aneuploidy frequency in preimplantation embryos is comparable to that in humans [27].

Many aneuploid cells or organisms die during embryogenesis. Aneuploidies are observed in 4–5% of all clinical pregnancies, the vast majority of which end in miscarriage [5,28]. Aneuploidies occur in approximately 50% of spontaneous abortions during the first trimester of pregnancy. Abnormal karyotypes are distributed in the following order: single autosomal trisomy (40.5%), triploidy (approximately 15%), and X monosomy (7.5%) [29]. In spontaneous abortions, trisomy has been observed on all chromosomes, but trisomy on chromosomes 2, 13, 15 16, 18, 21, and 22 is the most frequent [28]. Often, a mosaic karyotype is observed in spontaneous abortions [30,31]. In humans, embryos with trisomy 13, 18, and 21 and sex chromosome aneuploidy survive until birth.

## 3. DNA Methylation and Demethylation, Mechanisms and Key Players

Before discussing the relationship between DNA methylation and aneuploidy in ontogenesis, it is necessary to consider the main features of DNA methylation and introduce the main participants in this complex process.

DNA methylation, the addition of a methyl group to position 5 of cytosine to form 5mC, is the main form of DNA modification in the mammalian genome (Figure 1). The distribution of 5mC in the genome is bimodal. In general, repetitive sequences such as transposons, centromeric, and pericentric repeats are highly methylated, and the gene bodies of highly expressed genes are also methylated, whereas CpG islands, i.e., GC-rich sequences of 500–2000 bp, often present in promoter regions, are usually devoid of methylation [32]. DNA methylation is important for mammalian development and is involved in various biological processes, including transcription regulation, transposon silencing, X chromosome inactivation, and genomic imprinting [33]. Aberrant patterns of DNA methylation and mutations of genes encoding enzymes or DNA methylation regulators are associated with developmental disorders and cancer [34,35]. The main players in the process of establishing and maintaining DNA methylation are undoubtedly DNA methyltransferases (DNMTs) (Figure 1). Several different DNMTs are currently known. De novo methylation is performed by two methyltransferases, DNMT3A and DNMT3B, which have a similar structure, including a C-terminal catalytic domain with conservative catalytic motifs and an N-terminal regulatory domain [36]. These enzymes are highly expressed in blastocysts, and their expression is suppressed in differentiated cells. DNMT3A and DNMT3B DNA substrates contain unmethylated and hemimethylated CpG sites [37]. DNMT1 is a maintaining methyltransferase, prefers DNA substrates containing hemimethylated CpG sites, and is responsible for maintaining DNA methylation during cell division [38].

Another key player in DNA methylation is the UHRF1 protein, also known in humans as ICBP90 (Figure 1). This ubiquitin-like protein plays an important role in cellular events such as proliferation, cell cycle regulation, DNA damage control, and carcinogenesis [39,40,41]. UHRF1 can bind to methylated CpG nucleotide sequences when recruiting DNMT1, which maintains the methylation status during DNA replication [42]. UHRF1 and DNMT1 form a complex and are localized at replication sites and heterochromatin, and in the absence of UHRF1, DNMT1 is not localized in these regions. In addition, UHRF1 can also bind to DNMT3A and DNMT3B [43].

DNA demethylation can occur passively or actively (Figure 1). Since DNA methylation patterns are maintained by the DNMT1 and UHRF1 complex during replication, a functional deficiency of maintaining methylation can lead to a replication-dependent dilution of 5mC, known as passive DNA demethylation. Active DNA demethylation refers to an enzymatic process in which 5mC bases, also in their oxidized forms, are replaced by unmodified cytosines regardless of DNA replication. This transformation depends on enzymes in the ten-eleven translocation protein (TET) family that consists of three proteins, namely TET1, TET2, and TET3, which catalyze the sequential oxidation of 5mC to 5-hydroxymethylcytosine (5hmC), 5-formylcytosine (5fC), and 5-carboxylcytosine (5caC) [44,45]. 5fC and 5caC can be recognized and excised from DNA by thymine DNA glycosylase (TDG) [44,46,47]. The remaining abasic site can then be restored by base excision repair. This process leads to the replacement of 5mC with unmodified cytosine, completing “active” demethylation. It is important to note that passive 5mC dilution is also involved in TET-mediated DNA demethylation. Since oxidized 5mC derivatives are poorly recognized by DNMT1 [48], these tags may be lost during DNA replication.

Thus, DNA methylation is a dynamic modification, which can be changed in both directions. This plasticity is necessary for the basic functions of DNA methylation: (1) epigenetic control of gene expression; (2) limiting the activity of transposons; (3) maintaining a specific state of chromatin in individual regions of chromosomes (e.g., centromeres and pericentromeres); and (4) ensuring the stability of information in the germ line genome. These functions are provided by highly coordinated waves of epigenetic reprogramming of the genome during ontogenesis. As shown below, they may also partly underlie the relationship between DNA methylation and aneuploidy.

## 4. DNA Methylation and Aneuploidy in Gametogenesis

Although the events that occur in gametogenesis are mostly common to both sexes, the time intervals of gametogenesis in men and women are completely different. MI takes years in females but less than 14 days in male mice [49]. Oogenesis consists of two stages. First, gonocytes migrate to the area of the genital ridge after the second week of development and turn into oogonia. Oogonia actively divide via mitosis until five months of pregnancy, during which they increase in number from 2000 to approximately seven million. At the end of the proliferation phase, the active atresia of oogonia starts. As early as three months of pregnancy, most oogonia complete mitotic divisions, are converted into oocytes, enter prophase I and reach diplotene. Starting from the seventh month of pregnancy, oogenesis stops at dictyotene. The completion of meiosis (maturation of oocytes to obtain fertilized gametes) does not occur until the egg is fertilized during the ovarian cycle. Fertilization may occur in any cycle between menarche and menopause; thus, in humans, oocytes may remain in the dictyotene of MI between ~13 and ~51 years. Even in mice, the dictyotene stage lasts up to two years [50].

When forming male gonads, PGCs are surrounded by coelomic epithelial cells and form sex cords. Spermatogonia remain in this state until puberty. Spermatogenesis in humans lasts 72 days. During this time, two meiotic divisions take place, and the spermatid matures to a spermatozoon. During maturation, significant changes occur in the cell nucleus. DNA loses its nucleosomal organization, histone proteins are replaced with basic proteins rich in arginine and protamine, and DNA condensation is maximized.

Epigenetic reprogramming in the germ line involves somatic methylation patterns erasing in primary germ cells and then establishing sex-specific germ cell methylation patterns, including methylated tags in imprinting control areas. Immediately after separation of the germ line, the methylation profiles of PGCs are similar to those of somatic blastomeres. PGCs then undergo DNA demethylation in two different phases. This results in a decrease in methylation throughout the genome (Figure 2). The timing of the first wave of reprogramming has been well studied in mice. The first phase occurs during PGC proliferation and migration and involves passive demethylation [51]. The second phase leads to the demethylation of specific loci, including imprinted genes, CG dinucleotides in islands of the inactive X chromosome in females, and germ line-specific and meiosis-specific genes. Active demethylation is mainly responsible for removing methylated tags at these loci, which are protected from passive demethylation in early PGCs [9].

The establishment of specific markers of germ cell methylation after PGC reprogramming occurs at different times and in different cell environments in males and females, which ultimately leads to sperm- and oocyte-specific patterns [52,53]. In female germ cells, DNA methylation levels remain low, while male PGCs have already reached 50% of global DNA methylation [54]. The methylation patterns of male germ cells are fully established at birth and then persist for many mitotic division cycles before the cells enter meiosis [55,56]. The levels of DNA methylation in oocytes remain low before birth and after birth, whereas repeated methylation occurs in the oocyte growth phase [54]. Recent studies in humans indicate that, in general, the processes of the first wave of epigenetic reprogramming are similar to those in mice [57].

Regardless of the stage in which it originates, aneuploidy occurs as a result of two main events: chromosome nondisjunction and anaphase lagging. Most of the aneuploidies that occur in gametogenesis are the result of chromosome nondisjunction [58].

Meiotic errors can occur at several stages. First, erroneous chromosome segregation during and after meiosis I (MI) is often the result of either inability to conjugate or incorrect crossing over. During prophase I, homologous chromosomes conjugate and form bivalents for subsequent crossing over. If there is no normal conjugation of homologs, the chromosomes do not form bivalents and remain univalent, which eventually leads to errors in the segregation of homologous chromosomes. Normal crossing over occurs within the arm of the chromosome, but in the case where the crossing over regions are localized either near the telomere or in the pericentromeric regions, incorrect homolog segregation also occurs [59]. 

One of the effects of DNA methylation on maintaining the stability of the germ cell genome is its supposed ability to control crossing over and recombination in the centromere during meiosis and mitosis. Centromeres are known as “cold spots” for recombination. When crossovers occur in pericentromeric regions, chromatid segregation defects during MII have been observed in various species, including humans, *Drosophila melanogaster*, plants, and *Saccharomyces cerevisiae* [60]. Interestingly, even single gene conversion near centromeres is associated with 60% of segregation errors in meiosis I in *S. cerevisiae*, mainly due to the premature separation of sister chromatids [61]. Thus, crossovers must be inhibited in the centromere region to avoid destruction and loss of chromosomes after loss of cohesion between sister chromatids and subsequent chromosome loss or non-disjunction. In this context, DNA methylation can serve as a protective factor. In plants, the DNA methylation on centromeres is important for suppressing crossing over in these regions [60,62]. However, it is not known whether this mechanism exists in animals.

Second, the reason for chromosome nondisjunction is often the premature separation of sister chromatids in MI and MII due to a loss of cohesion between chromatids [13,63,64,65,66]. This effect may be the result of aberrant DNA methylation in genes involved in ensuring the stability of cohesion between sister chromatids. To date, several such potential genes are known.

Crossing over “slippage” was observed in mice with Smc1β cohesin complex gene knockout. That is, while DNA mismatch repair protein MLH1 foci were located normally along with synaptonemal complexes on condensed chromosomes, chiasma was located closer to telomeres [67]. Indeed, the connections between homologous chromosomes are often lost during diakinesis. It is important to note that the effect was compounded by maternal age: in older women, the chromosomal connections simply disappeared, and the chromosomes were present as separate sister chromatids. Another gene whose malfunctioning leads to meiotic aneuploidy is *MEI1*. Its product is involved in the formation of double-stranded DNA breaks in MI during the induction of recombination. Gene mutations lead to the abnormal segregation of all chromosomes in MI into the first polar body [68]. When such oocytes are fertilized with two spermatozoa or one diploid spermatozoon, recurrent hydatidiform mole-3 occurs (OMIM 618431). *MEI1* gene expression is also regulated by DNA methylation, at least in cattle [69]. One of the important genes involved in meiotic recombination is the *RAD51C* gene. Mouse oocytes with a hypomorphic mutation in this gene progressed normally to metaphase I after superovulation but showed premature sister chromatid separation, aneuploidy, and chromatid breaks in metaphase II [70]. Constitutive epimutations in the *RAD51C* gene with a frequency of 1.3% were detected in DNA from the whole blood of patients with familial forms of breast and ovarian cancer [71]. These data suggest that epimutations likely occurred either in the oogonia or in blastomeres at the cleavage stage. Some embryos with low levels of mosaicism can be implanted and live to later pregnancy but then undergo spontaneous abortion.

In addition, an important cause of mitotic and meiotic aneuploidies is impairment of the spindle assembly checkpoint (SAC). SAC malfunction in meiosis can lead to the formation of aneuploid gametes, which in the vast majority of cases are incompatible with subsequent embryonic development, presumably because additional chromosome leads to a very large gene imbalance [50]. Central to SAC function is the kinetochore and anaphase-promoting complex, the downstream target of the SAC, which irreversibly promotes the progression of cells from metaphase to anaphase. The SAC is able to control the progression of the cell cycle, allowing progress to anaphase only when all chromosomes have formed the correct attachments to the spindle [50].

Thus, the first wave of genome methylation reprogramming coincides with the increase in the frequency of aneuploidies in gametogenesis (Figure 2). Given that demethylation leads to simultaneous changes in the expression level of thousands of genes, chromatin remodeling throughout the genome and reducing the control over mobile genetic elements, even slight variation in this process can lead to increase in the incidence of aneuploidy during meiotic divisions.

## 5. DNA Methylation and Aneuploidy at the Cleavage Stage

The second wave of global epigenetic reprogramming occurs during early embryogenesis and begins immediately after fertilization (Figure 2). The newly formed embryo undergoes marked global DNA demethylation such that, by the time the early blastocyst stage (32–64 cells) is reached, methylation is at its lowest level [72]. Most of the genome is demethylated, with the exception of evolutionarily young CpG-rich transposons and imprinted regions. The paternal and maternal genomes follow different DNA demethylation kinetics. The rate of demethylation of the paternal genome is much higher than that of the maternal genome at the zygote stage.

The reprogramming of the embryonic genome occurs in three stages. The first wave is complete within the first 10–12 h after fertilization. The average level of DNA methylation in the paternal genome decreases from 82% in the sperm to 53% in the male pronucleus. At the same time, the average level of DNA methylation decreases from 55% in the mature oocyte to 51% in the maternal pronucleus. After the two-cell stage, residual methylation in the paternal genome (on average 15% at the two-cell stage) is lower than that in the maternal genome (on average 23% at the two-cell stage) throughout preimplantation development [73]. The maternal genome undergoes passive DNA demethylation during several cycles of DNA replication, while the paternal genome is actively demethylated. Paternal 5mC is actively converted to 5hmC (and possibly further to 5fC and 5caC), which then undergoes replication-mediated dilution during subsequent cleavage [74,75]. DNMT1 shows a very limited affinity for oxidized 5mC derivatives [48] and is usually excluded from the nucleus of preimplantation embryos [76,77]. During this step, the enhancer regions and gene bodies are demethylated.

The second step of global demethylation occurs from the late zygote to the two-cell stage, with the level of DNA methylation decreasing from 50% to 40%. The third step lasts from the eight-cell stage to the morula stage, while the methylation level decreases from 47% to 35%. In these stages, intron regions and short interspersed nuclear elements (SINEs) undergo strong demethylation, especially those in the evolutionarily young subfamily of Alu elements [73].

This wave of DNA demethylation coincides with steep increase in the incidence of aneuploidy during the first three cell divisions of the cleavage stage (Figure 2). However, data on the association of methylation with the induction of aneuploidy in the early stages of human development are very limited and contradictory. It was shown that low-quality blastocysts differed from high-quality blastocysts in the presence of certain DMRs with an average length of 2 kb, localized in both CpG islands and gene shores. These regions are enriched with genes that regulate pathways critical for embryonic development, including the cell cycle, DNA metabolism and modification, and chromosome localization [78]. However, when analyzing aneuploid blastocysts, the authors did not find significant changes in the methylome in aneuploid blastocysts, both high quality and low quality. In a study by Zhu and coauthors, it was shown that the genome-wide level of methylation in blastomeres with monosomies was increased compared to euploid blastomeres, and blastomeres with trisomy had a reduced level of methylation both in the trophoblast and in the inner cell mass [73]. However, since the researchers were only interested in the DNA methylation profile of euploid blastocysts, they did not consider the details of the differential DNA methylation of embryos with aneuploidy. In an earlier study, no differences were found in the DNA methylation profile of blastocysts with trisomy from euploid blastocysts, but in cases of monosomies, individual chromosomes with monosomy were hypermethylated. A decrease in DNA methylation has been recorded in several genes that are necessary for implantation and normal embryo development [79]. Moreover, along with a decrease in the level of gene methylation on chromosomes with monosomy, there was also a decrease in the transcription level of DNA methyltransferase genes. However, it remains unclear whether these phenomena are the cause or consequence of chromosome loss.

The first three divisions of cleavage occur under the control of parental factors, since the embryo genome has not yet been activated. This indicates the potential association of maternal and paternal factors with the epigenetic reprogramming and the origin of aneuploidy [80]. For example, maternal UHRF1 has been shown to control DNA methylation and histone modification in mouse oocytes and plays an important role in the development of preimplantation embryos (Figure 1). Conditional deletion of UHRF1 in growing oocytes led to the disturbance of spindle and chromosome separation and increased DNA damage. These defects corresponded to the low quality of oocytes. Although mouse experiments both in vivo and in vitro showed that UHRF1-deficient oocytes could be fertilized, embryos derived from these oocytes could not reach the blastocyst stage [81].

Mosaicism origin may also be associated with sperm defects. During the spermatogenesis process, sperm chromatin undergoes a specific epigenetic mechanism that involves a major structural revision of DNA packaging caused by the replacement of histones with protamines. However, some regions do not bind to protamines but maintain the nucleosomal type of chromatin organization. The histones in these nucleosomes carry epigenetic signatures that are transferred to the oocyte during fertilization, therefore affecting DNA accessibility for transcription factors and regulating gene expression in the early embryo, as well as being passed on to offspring. These remaining nucleosomes are not randomly distributed in the paternal genome. The regions containing LINEs and SINEs are also enriched with nucleosomes [82]. In addition, regions free of DNA methylation in the early embryo correspond to regions rich in nucleosomes in the chromatin of spermatozoa [83].

Aberrant methylation of such areas can potentially lead to the impaired activation of the embryo’s genome and to aneuploidy. Delays in the decondensation of chromatin from the sperm apical region impeded the progression of the first mitotic division of the zygote [84]. Aberrant methylation of sperm DNA and histones has been reported in infertile men [85]. In a study of 63 men whose female partners participated in an IVF program, sperm DNA hypermethylation across the entire genome was associated with pregnancy failure [86]. Another study in which the sperm global DNA methylation index was measured in the genomes of IVF patients and normozoospermic fertile men showed that sperm DNA methylation patterns can predict embryo quality during IVF [87]. The researchers noted that, although “poor” embryogenesis (developmental delay or depletion at any point during embryonic development or poor implantation) cannot be attributed to several consecutively modified CG dinucleotides, the whole genome methylation profile in fathers of poor-quality embryos was significantly different from the norm. It has been reported that abnormalities in paternal sperm chromatin condensation correlate with chromosomal aneuploidies in embryos [88,89]. In men with oligoasthenoteratozoospermia, a condition characterized by altered epigenetic profiles in sperm [90], there was an increased level of disomy, diploidy, and nullisomy compared to their fertile counterparts, and the level of aneuploid embryos is also increased [91].

Several studies have also examined the effect of increased paternal age on the level of sperm aneuploidy [92,93]. Two recent studies note that compared with men less than 50 years of age, men over 50 years of age in IVF/ICSI cycles had significantly more sperm with damaged DNA, a lower rate of blastocyst development, a higher rate of aneuploidy, and a significantly higher number of embryos with trisomies [92,93]. Increased paternal age leads to several changes in the male endocrine system and reproductive phenotypes (changes in testicular morphology and volume, changes in sperm production and characteristics, and a significant increase in sperm DNA fragmentation) [92,94]. In addition, the level of DNA methylation in spermatozoa increases with age in human and bull sperm [95,96]. This, combined with the accumulation of DNA damage over many years and a decrease in the ability of germ cells to repair this damage, may lead to a decrease in the integrity of the sperm genome, which leads to the formation of aneuploid spermatozoa and an increase in aneuploidy in embryos [97].

Simultaneously, with DNA demethylation, de novo DNA methylation begins at the preimplantation stages of development, which occurs in two stages (Figure 2). The first stage starts from the early male pronuclear stage to the mid-pronuclear stage, and the second occurs from the four-cell stage to the eight-cell stage [73]. De novo-methylated sites are mostly located in repetitive sequences, namely long interspersed nuclear elements (LINEs), long terminal repeats (LTR), and Alu repeats. This process is probably necessary to suppress their transcriptional activity to avoid genome instability.

After implantation, the embryonic cells acquire tissue-specific DNA methylation, and the global level of genome methylation increases (Figure 2). In contrast, in extraembryonic tissues, including trophoblasts and extraembryonic mesoderm, the level of DNA methylation remains low [98]. This coincides with the high level of aneuploidy observed in trophoblast cells [99]. After birth, the level of DNA methylation remains constant and then changes, including global hypomethylation and region-specific hypermethylation, with aging and age-related disease development [100]. At the same time, there is an increase in the aneuploidy frequency in cells during both normal aging [101] and various pathologies, including malignant tumors [102] and neurodegenerative diseases [103].

Thus, the genome is subjected to two reprogramming processes: one at the stage of primary germ cells and the other at the cleavage stage. Both of these stages are characterized by an increased aneuploidy frequency. The concept of an “epigenetic landscape” implies the presence of a stable epigenetic background, so-called “valleys”, which are each type of cell in the ontogenesis of the organism, and ridges separating these valleys [104]. It can be assumed that once removed from the state of epigenetic equilibrium, the system becomes more unstable until it reaches a new “epigenetic valley”. This instability can be expressed in an increase in the frequency of structural and numerical chromosomal abnormalities, as well as, possibly, epigenetic disorders. An example is an increase in the frequency of aneuploidy at the stage of epigenetic reprogramming in the zygote and during the initial cleavage, accompanied by a sharp decrease in the global level of genome methylation (Figure 2). Similar processes seem to occur when reprogramming somatic cells to produce induced pluripotent cells. Finally, a similar, although much longer in time and largely spontaneous, coupling of changes in the epigenetic background of cells and the stability of their genome occurs during aging [101].

## 6. Potential Mechanisms for the Relationship between DNA Methylation and Aneuploidy

The observed mirrored dynamics of the level of DNA methylation and the frequency of aneuploidy in ontogenesis indicates possible relationship between the processes of epigenetic reprogramming and ensuring the stability of the genome. On the one hand, gain or loss of chromosomes can affect the final epigenetic status of the cell or the entire organism; on the other hand, errors in the DNA methylation reprogramming of certain regions of the genome can induce aneuploidy (Figure 3). The effect of aneuploidy on the DNA methylation profile is described in more details, while the opposite effects are still poorly understood.

### 6.1. Influence of Aneuploidy on Certain Chromosomes on the Level of DNA Methylation

The main effect of aneuploidy is the change of the expression of thousands of genes located on the extra or lost chromosome, as well as in other parts of the genome. These may include genes whose products play important roles in establishing or maintaining DNA methylation. In addition, the cell responds to aneuploidy (in particular, to the extra chromosome) by epigenetically compensating an increased gene dosage by increasing the level of DNA methylation in promoters of these genes. These processes can lead to epigenome imbalance, when the entire DNA methylation profile changes throughout the genome (Figure 3).

An interesting model for studying the relationship between DNA methylation and aneuploidy induction is X chromosome monosomy. X chromosome monosomy is the most common genetic abnormality in humans, as it is present in approximately 2% of all human pregnancies, although 99% of these pregnancies end in spontaneous miscarriage. Postnatally, monosomy X leads to Turner syndrome (TS) with clinical signs of typical dysmorphic stigmata, low stature, sexual infantilism, and renal, cardiac, skeletal, endocrine, and metabolic disorders. The syndrome is characterized by a variety of clinical manifestations, which may be the result not only of a decrease in the dosage of genes corresponding to the missing chromosome but also of epigenetic changes [105]. When comparing the genome methylation profile in peripheral blood samples of TS patients, normal men 46, XY and women 46, XX, TS were shown to be associated with a large number of differentially methylated CG sites (more than 800 sites), most of which were hypomethylated [106,107]. DMRs were located throughout the genome, including autosomes, but were not located in CpG islands but in shores. These data are of considerable interest since the second X chromosome in women is always inactivated, and only 15–20% of the genes on the X chromosome avoid inactivation [108]. Most of these genes are located on the short arm of the X chromosome in the pseudoautosomal region and are duplicated on the Y chromosome in males. DMRs in TS patients were localized in genes that showed a strong enrichment of immune system processes, immune response, protective response to bacteria, cytokine production, and meiosis [106]. This global hypomethylation may be the cause of X chromosome instability. Interestingly, studies on the methylation profile of peripheral blood samples in patients with Kleinfelter syndrome indicate that the number of DMRs in Kleinfelter syndrome patients is five times less than that in TS patients [106].

Studies of the effect of aneuploidy on the DNA methylation profile were also conducted on trisomy 21, which is the cause of Down syndrome (DS). mRNA profiling in cells and tissues with trisomy 21 revealed widespread changes in gene expression, mostly small in size, both for genes on chromosome 21 and for large groups of genes on other chromosomes [109]. An analysis of the genome-wide methylation in peripheral blood lymphocytes revealed gene-specific hypermethylation. Aberrant methylation concerned genes necessary for the normal development and functioning of lymphocytes, as well as genes associated with the immune response [110]. Reduced-representation bisulfite sequencing (RRBS) at single-nucleotide resolution in the placenta revealed global DNA hypermethylation in all autosomes in DS samples [109]. The predominance of hypermethylation over hypomethylation in DS was observed in all genomic regions (promoters, intragenic regions, intergenic regions, and transcription termination regions). This predominance of hypermethylation was most pronounced in promoter regions, particularly promoters that overlap with CpG islands. In total, 589 genes in all autosomes were hypermethylated in promoters in DS with the enrichment of genes involved in the physiology and activity of neurons and the immune response. Additionally, it was shown that the expression of the TET family genes involved in DNA demethylation was significantly reduced in trisomy 21 samples, all of which were suppressed in the DS samples studied [109]. However, an analysis of the placenta methylation profile of embryos with trisomy of chromosome 21 in the first trimester of pregnancy using microarray technologies did not show significant differences compared to euploid placentas. Only three differentially methylated CpG sites (DMS) were identified [111]. Despite this, embryos with trisomy 21 die very often at this stage of development [29].

The study of the effect of aneuploidy on other chromosomes on the DNA methylation profile mainly concerned embryos at various stages of development. Trisomy of chromosome 16 is the most common aneuploidy in spontaneous abortions during the first trimester of pregnancy. Hypermethylation of multiple CG sites was also detected in chorionic villous cytotrophoblasts in spontaneous abortions of the first trimester of pregnancy with trisomy of chromosome 16. Hypermethylated sites are responsible for cell development, cell adhesion, immune response, and stimulus response [112]. Moreover, the DNA methylation profile of placentas with trisomy 16 in the first trimester of pregnancy overlapped with that of third-trimester placentas with early-onset preeclampsia [113]. It is likely that abnormal chorion development may be critical at this stage of development.

In placentas with trisomies 13 and 18, large differences were found compared to normal placentas [111]. In trisomy 13, 219 hypermethylated sites were found, which were localized on different autosomes and in different regions of genes. Differentially methylated genes performed various functions, and there was no significant enrichment in certain metabolic pathways. For trisomy 18, 217 hypermethylated CpGs were identified, which were also localized on various autosomes and were located mostly in CpG islands but not in gene promoters. After analyzing the functions of differentially methylated genes using GO enrichment analysis, it turned out that biological pathways associated with DNA binding and transcription factor binding coupled to RNA polymerase II-mediated transcription are significantly enriched. This significant number of differentially methylated genes in trisomy 13 and 18 compared to trisomy 21 is explained by the authors by the more severe phenotypes of trisomies 13 and 18 [111].

Thus, the extra or lost chromosome in cell significantly changes the DNA methylation profile throughout the genome. Such changes can lead either to decreasing the level of DNA methylation (monosomy X) or to the hypermethylation (trisomy 13, 16, 18, and 21).

### 6.2. Effects of DNA Methylation on the Incidence of Aneuploidy

How and when can aberrant DNA methylation disrupt normal chromosome/chromatid segregation and induce aneuploidy during early ontogenesis? These effects can be mediated through a variety of mechanisms associated with disturbances in the realization of the main functions of DNA methylation, discussed earlier. The disturbance of the epigenetic control of various genes expression, including cell cycle control genes and genes of DNA methyltransferases, can lead to genome instability and higher incidence of aneuploidy (Figure 3). Reduced control over the activity of transposons can lead to an increased probability of their transposition and disruption of many genes, which in turn can also lead to aneuploidy. Disturbed chromatin conformation in centromeres and pericentromeres is directly related to the occurrence of aneuploidy due to the important role of these regions in chromosome segregation. Finally, as discussed above, disruption of epigenetic reprogramming mechanisms involving maternal and paternal factors can lead to aneuploidy in germ line.

Aneuploidy can be induced due to the functional inactivation of genes that control the cell cycle or the number of centrosomes. Aberrant DNA methylation in these genes or their regulatory elements may be one of the reasons for functional inactivation (Figure 3). We previously detected hypermethylation of some tumor suppressor genes, *RB1* and *p14ARF*, in extraembryonic tissues of human miscarriages with mosaic aneuploidy [114,115]. It has been shown that mouse and human primary pRb-deficient fibroblasts are characterized by the occurrence of enumerated chromosomes [116]. In addition, pRb loss causes centrosome amplification and aneuploidy in Rb-knockout murine fibroblast lines [117]. Later, there was evidence that, in the case of DNMT1 depletion in almost diploid human tumor cells (HCT116), which are p14ARF-null, the cell cycle is not arrested, leading to genome instability [118].

DNA hypomethylation during both waves of genome reprogramming is accompanied by a limitation of the function of methyltransferases, which in itself can lead to genomic instability both in oogenesis and in early embryo development. This assumption is supported by experiments with model animals and cell lines with depleted DNMT functions. Mice with DNMT1 deficiency are predisposed to tumors as a result of genomic instability induction. Human cancer cell lines with DNMT1 and/or DNMT3b knockout showed large abnormalities in the number of chromosomes [119,120]. Dnmt3b^-/-^ mouse embryonic fibroblasts (MEFs) contained 40% more polyploid cells than wild-type MEFs (∼20%) and Dnmt3a^-/-^ MEFs (∼20%). Dnmt3a^-/-^ MEFs showed chromosomal abnormalities characterized by the presence of fused broken ends of chromosomes and anaphase bridges [121]. The expression of DNA methyltransferases is also associated with aneuploidy in embryogenesis: all four DNMT genes were downregulated in blastocysts with chromosome 15 monosomy compared to euploid blastocysts [79].

A possible reason for the effect of a decrease in the global genome methylation level on the frequency of aneuploidy may be cytosine hypomethylation in centromeric and pericentromeric repeat sequences [122,123,124], leading to abnormal kinetochore structure, misattachment of microtubules, and incorrect function of the spindle assembly checkpoint [125,126]. Human centromeric chromatin is generally defined by the presence of α-satellite DNA sequences consisting of tandem head-to-tail repeats of 171 bp-long AT-rich monomers, representing approximately 3% of the genome. These monomers are further organized into higher-order repeats (HORs), spanning 340 bp to 6 kb. Globally, the repetition of HORs creates large domains of centromeric DNA approximately 0.3–5 Mb in size. The unique repetitive nature of the centromeric DNA sequence can give the centromere a complex DNA topology, making it a potentially fragile region of the genome [60]. The sequences flanking centromeres, called pericentromeres, are less organized and more heterogeneous than centromeres. Pericentromeres are also repetitive, mainly due to divergent DNA monomers mixed with mobile elements such as LINEs and SINEs. The pericentromeric chromatin is tightly packed and must be strong enough to absorb the mechanical stress caused by pulling the spindle microtubules on the kinetochore.

A sharp decrease in the level of DNA methylation in these regions during the first wave of reprogramming can lead to numerous errors in the distribution of chromosomal material if we take into account that the entire pool of oogonia accumulates in a short period of reproduction in embryogenesis. In studies on the methylation status of PGCs in mice, it was shown that the level of satellite DNA methylation in these regions was significantly reduced compared to that in cumulus cells in both male and female gametogenesis [127]. The methylation levels of major and minor microsatellites in oocytes were 16.5% and 26% compared to 86.9% and 89% in cumulus cells [127]. The second wave of genome reprogramming that occurs after implantation also leads to the hypomethylation of the pericentromeric regions of heterochromatin. This is especially noticeable in the methylation patterns of large heterochromatin blocks on chromosomes 1, 9, and 16 in human preimplantation embryos. An analysis of metaphase chromosomes from blastomeres of human preimplantation embryos showed a hypomethylated state of pericentromeric heterochromatin of these chromosomes compared with the hypermethylated state of these sites in postimplantation embryonic tissues and in lymphocytes [128]. The decrease in pericentromeric methylation in human lymphocytes treated with 5-azacytidine (AZA), causing loss of DNA methylation, is associated with the erroneous segregation of chromosomes 1, 9, 15, 16, and Y and with increases in the frequency of micronucleus formation [129,130]. This cytidine analog induces a distinct lack of condensation in the heterochromatic regions of chromosomes 1 and 16, which have highly methylated regions of satellite 2 [130]. This observation suggests a link between DNA methylation loss and chromosome loss. Treatment of conditionally diploid cells with low doses of 5-aza-2′-deoxycytidine (DAC), which cause DNA hypomethylation without affecting cell viability, induced DNA methylation loss in the centromeric/pericentromeric region. There was a significant increase in the number of hypodiploid cells [131].

DNA methylation is widespread in the centromere, and the aberrant DNA methylation observed in some tumors correlates with aneuploidy and genome instability [60]. A decrease in pericentromeric methylation in human cells treated with AZA and DAC, which causes DNA methylation loss, is associated with erroneous chromosome segregation and increased micronucleus formation [129,130,131]. These cytidine analogs induce a distinct decrease in condensation in the heterochromatic regions of chromosomes.

Decreases in α-satellite and satellite II DNA methylation, which occurs, for example, in several forms of immunodeficiency, centromeric region instability, facial anomalies (ICF) syndrome (resulting from mutations in the *DNMT3B, ZBTB24, CDCA7,* and *HELLS* genes), lead to the synthesis of a large number of transcripts of retrotransposons, resulting in retrotransposition and insertion in new sites in the genome. Hypomethylation of the LINE-1 retrotransposon is also associated with aneuploidy (Figure 3). During oocyte development, only a limited portion of the original fetal oocyte pool survives until postpartum follicle growth and ovulation; most developing oocytes are lost as a result of programmed cell death during differentiation and maturation. After the seventh month of pregnancy and before birth, fetal oocyte attrition occurs, resulting in the elimination of ~80% of the original pool of human oocytes by the time of birth. This process is not unique to humans; it has been observed in primates and widely documented in several rodent species. Remodeling the DNA methylation of an embryonic germ line creates a window of opportunity for the expression of retrotransposons, such as LINE-1, whose intact and mutated copies make up ~17% of the mammalian genome [132].

These transposable elements can reproduce in the genome using a reverse transcription mechanism. The main mechanism through which the activity of these elements is repressed is DNA methylation of the promoter region and PIWI-interacting RNAs (piRNAs) [133]. LINEs contain strong, usually hypermethylated promoters, and the decreased methylation of these promoters can notably increase transcription and retrotransposition [134]. Observations in some mammalian species have shown that retrotransposons can act as an integral component of centromeres. In primates, LINE-1 retrotransposon insertions are located between sequences of monomeric satellites throughout the pericentromeric region of the X chromosome [135,136]. The oldest of these insertions are located more distally, and the younger and more active ones are closer to the centromeric cortex. These new elements differ slightly in their sequence and can maintain the activity of their promoters, allowing for the transcription of nearby sequences. In addition, the enrichment of the genome sequence with LINE-1 retrotransposons was associated with the formation of neocentromeres [137,138,139]. These results indicate the possible role of LINE-1 in initiating centromere formation. Given the noted structural and functional role of the LINE-1 retrotransposon in the formation of centromeres and kinetochores, changes in its transcriptional activity may lead to the impairment of centromeric chromatin conformation and to errors in chromosome segregation during cell division. This change in the expression of centromeric LINE-1 can potentially be observed with a decrease in the global genome methylation level (including LINE-1), leading, in turn, to an increase in the frequency of aneuploidy [122,123,124].

Retrotransposon derepression is also associated with the coactivation of neighboring genes or chimeric transcripts consisting of retrotransposable elements and host genes [140]. In oocytes and two-cell embryos, this indirect retrotransposon-mediated gene regulation may be crucial for successful development [140,141]. In a study on the role of LINE-1 retrotransposon in fetal mouse oocytes, Malki and coauthors showed that the expression of LINE-1 elements after their activation during epigenetic reprogramming may be harmful to oocyte viability and induce aneuploidy [142]. A 2.4-fold increase in the expression of LINE-1 in female mice with knockout of the Mael gene, the product of which suppresses transposon expression, led to a three-fold decrease in the number of fetal oocytes, and the introduction of azidothymidine, a reverse transcriptase inhibitor, into pregnant Mael-null females had an immediate positive effect on the dynamics of fetal oocyte attrition [142]. A four-fold increase in embryonic oocytes in MI with the incomplete synapse of homologous chromosomes was shown in Mael-null females [142]. These data suggest that increased LINE-1 expression may interfere with crossover formation, leading to errors in meiotic chromosome segregation in the adult ovary.

Thus, several possible mechanisms of DNA methylation influence on the incidence of aneuploidy are currently known, which are associated with disturbance of the main functions of epigenetic regulation. However, most results are limited to mouse models or cell cultures, which imposes certain restrictions on their extrapolation to the generation of aneuploidy in human germ cells and embryogenesis.

## 7. Conclusions

Although the connection between aneuploidy induction and methylation reprogramming in early mammalian ontogenesis seems quite possible due to the obvious spatiotemporal coincidences of these processes, there is almost no direct experimental evidence confirming this relationship. The rapid demethylation of PGCs can itself induce aneuploidy, but the difference in the frequency of aneuploidy in male and female germ cells is not explicable from these positions. The probability of the accumulation of DNA methylation errors and the spread of mutations occurring due to the spontaneous deamination of 5hmC during this long period of cell replication and division is much higher in men than in women [72]. Moreover, the frequency of aneuploidy in oocytes is an order of magnitude higher than in spermatozoa. On the other hand, it can be assumed that the stricter SAC mechanisms in male germ cells do not allow the aneuploid cell to continue meiosis and turn into spermatozoa.

The first cleavage divisions are also characterized by an increase in the frequency of aneuploidy and by the reprogramming of the genome in the absence of mitotic checkpoints of the embryonic genome. The speed and complex nature of the processes occurring at these stages is such that we can only guess what an imbalance of DNA methylation may contribute to incorrect chromosome segregation at this stage—the hypomethylation of repetitive DNA sequences or insufficient removal of methylation in genes that control the cell cycle. However, if we take into account the fact that blastocysts with an aneuploidy frequency of up to 60% can reach implantation [143], perhaps due to an aberrant checkpoint, then the aberrant methylation of cell cycle control genes is very likely.

We can conclude that the incidence of aneuploidy and the level of DNA methylation in the human genome seem to be mirrored interfering features in early human embryo development. It is possible that aneuploidy and DNA methylation, being in reversed phase during critical periods of ontogenesis, limit the genome instability in a certain frame, providing possibility of individual development. However, there are very limited data about the cause-and-effect relationship between aneuploidy and methylation in early human ontogenesis, and the available data are very contradictory. It is likely that this is just the beginning, and the development of model systems including in vitro meiosis will allow us to comprehensively study the relationship of these important processes and will provide new opportunities.

## Figures and Tables

**Figure 1 genes-11-01084-f001:**
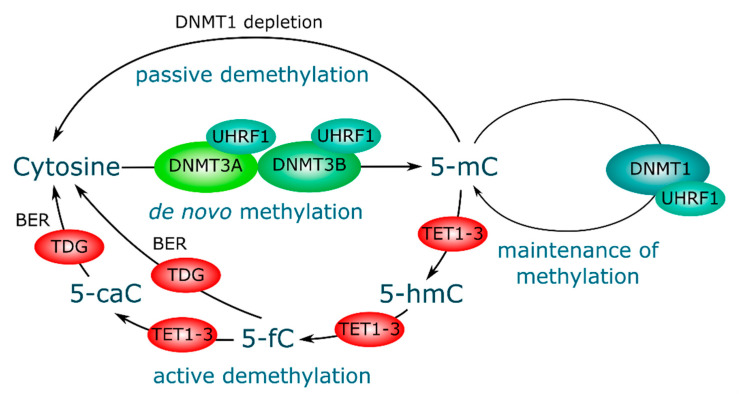
Mechanisms and key players of DNA methylation and demethylation.

**Figure 2 genes-11-01084-f002:**
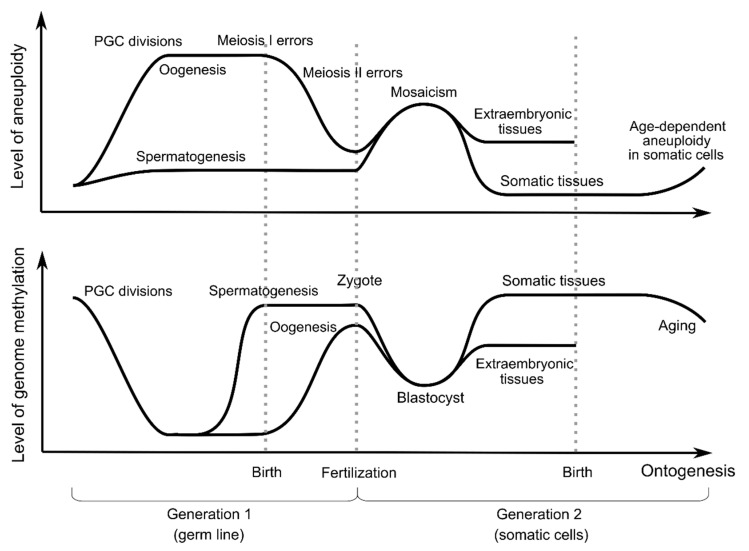
Mirrored dynamics of the levels of genome methylation and aneuploidy in human cells during ontogenesis.

**Figure 3 genes-11-01084-f003:**
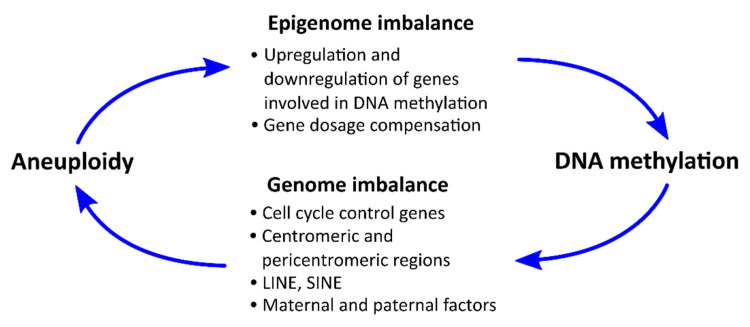
Potential relationships between DNA methylation and aneuploidy.

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
