# Peer review of "Aneuploidy and DNA Methylation as Mirrored Features of Early Human Embryo Development"

_genes, 2020, doi:10.3390/genes11091084_

Round 1
Reviewer 1 Report
This is an excellent review article that comprehensively summaries evidence, albeit mostly substantial, linking aneuploidy to DNA reprogramming during development. I have only several minor suggestions.
- Lines 579-580, under subtitle 15, there is only one sentence in the section, which looks odd. The authors should either expand the section or delete the section and add that one sentence to another section.
- Lines 396-401, the information in the last paragraph of the section appears to be repetitive to that described in lines 386-395.
- DNMT3A and DNMT3B were used in some places, and DNMT3a and DNMT3b were used in other places. I'd suggest DNMT3A and DNMT3B be used throughout the manuscript.
- Line 342, regarding the phenotype of DNMT1 KO in human cancer cell lines, the paper by Chen et al. (Nat Genet 2007) should also be cited.
- Some typos and errors need to be corrected. Line 126, "until the egg is not fertilized..." should be "until the egg is fertilized...". Line 134, "acidic" should be "basic". Line 347, DNMT was misspelled as DNTM. Line 564, disomy was misspelled as dysomy.
Author Response
Dear Reviewer
The authors are sincerely grateful for your special attitude to the work and critical comments. In accordance with them, the design of the manuscript has been significantly revised. The authors hope that this improvement has increased the clarity of the idea of the work.
Please, find below the authors' responses to each of the reviewers' comments.
This is an excellent review article that comprehensively summaries evidence, albeit mostly substantial, linking aneuploidy to DNA reprogramming during development. I have only several minor suggestions.
Thank you for the excellent evaluation of our review article!
Point 1: Lines 579-580, under subtitle 15, there is only one sentence in the section, which looks odd. The authors should either expand the section or delete the section and add that one sentence to another section.
Response 1: The authors apologize for the incorrect numbering of chapters in the manuscript, which caused misunderstanding. The Chapter 15 was intended as a unifying one for subsequent chapters (16-19). In the revised version of the manuscript, chapters 15-19 have been merged into one section 6.1.
Point 2: Lines 396-401, the information in the last paragraph of the section appears to be repetitive to that described in lines 386-395.
Response 2: The last paragraph in this section has been deleted.
Point 3: DNMT3A and DNMT3B were used in some places, and DNMT3a and DNMT3b were used in other places. I'd suggest DNMT3A and DNMT3B be used throughout the manuscript.
Response 3: Line 127 in revised version, “DNMT3a” and “DNMT3b” has been replaced by “DNMT3A” and “DNMT3B”.
Point 4: Line 342, regarding the phenotype of DNMT1 KO in human cancer cell lines, the paper by Chen et al. (Nat Genet 2007) should also be cited.
Response 4: Line 489 in revised version, Chen et al. (Nat Genet 2007) has been cited.
Point 5: Some typos and errors need to be corrected. Line 126, "until the egg is not fertilized..." should be "until the egg is fertilized...". Line 134, "acidic" should be "basic". Line 347, DNMT was misspelled as DNTM. Line 564, disomy was misspelled as dysomy.
Response 5: All proposed changes have been made to the manuscript:
line 126 (line 159 in revised version), "until the egg is not fertilized..." has been changed to "until the egg is fertilized..."
line 134 (line 167 in revised version), "acidic" has been changed to "basic"
line 347 (line 493 in revised version), “DNTM” has been changed to “DNMT”
line 564 (line 333 in revised version), “dysomy” has been changed to “disomy”

Reviewer 2 Report
In this review authors go through the relationship between DNA methylation and aneuploidy in human and mammalian ontogenesis. This is a especially difficult topic to approach, so I congratulate the authors for the work done. Nevertheless, in my humble opinion, there are some points that should be addressed to consider this review for publication.
Major comments:
- Even though in the abstract authors say that they are going to consider the relationship between DNA methylation and aneuploidy in human and mammalian ontogenesis, the title is focus on human embryos. In some parts of the review, like in 6 and 7, there is almost no reference to human, only to animal models. Maybe the information is not available, but this should be stated.
- I think that if authors added some figures explaining the methylation processes the manuscript will be easier to understand and more dynamic (for example in point 3).
- In figure 1, in the level of genome methylation graph, I am sure is obvious but I do not understand the birth stages, could you explain this to me, please?
- Point 8 is confusing to me, I am not sure if authors are talking about aneuploidies in somatic or embryonic cells.
In summary: I am aware that this topic is very complicated and hard to explain. Here relays the difficulty of writing a good and understandable review. Authors have worked very hard and they have recollected a huge amount of information. Nevertheless, in my opinion the order of the information could be improved, to facilitate the reading and understanding. For example, the review should start with the general aspects of methylation (processes, mutation and so on) and then link this information with the generation of aneuploidies in gametes and embryos. It is very interesting to know about the carcinogenesis and animal models, but I think the main point is not clear once you read all the manuscript. I consider that rewriting is necessary to obtain a fluent explanation of the generation of aneuploidies due to methylation issues. In the end, the reader should have to be able to follow the manuscript like a story, not like paragraphs of information that are appearing, and you do not know exactly how to associate them with the rest of the information.
Minor comments:
Line 65: AMA starts 35-38 y-old not 33, and with this the decreasing in the fertility curve.
Line 82: The percentages of mosaic keryotypes in humans seems to high. Could authors make sure this data is correct? And, what about mosaicism in preimplantation embryos?
Line 116: reference is missing.
470: Drosophila should be in italics, and better to write the complete name of the specie D. melanogaster. Does the term yeast only refer to S. cerevisiae or to yeast in general? because if this is the case, it should be stated here.
DNA methylation crossing over: in order to make this
Line 508: reference? This part is so short that maybe could be added to other point of the manuscript.
Author Response
Dear Reviewer
The authors are sincerely grateful for your special attitude to the work and critical comments. In accordance with them, the design of the manuscript has been significantly revised. The authors hope that this improvement has increased the clarity of the idea of the work.
Please, find below the authors' responses to each of the reviewers' comments.
This is an excellent review article that comprehensively summaries evidence, albeit mostly substantial, linking aneuploidy to DNA reprogramming during development. I have only several minor suggestions.
Thank you for the excellent evaluation of our review article!
In this review authors go through the relationship between DNA methylation and aneuploidy in human and mammalian ontogenesis. This is a especially difficult topic to approach, so I congratulate the authors for the work done. Nevertheless, in my humble opinion, there are some points that should be addressed to consider this review for publication.
Thank you for the excellent evaluation of our review!
Major comments:
Point 1: Even though in the abstract authors say that they are going to consider the relationship between DNA methylation and aneuploidy in human and mammalian ontogenesis, the title is focus on human embryos. In some parts of the review, like in 6 and 7, there is almost no reference to human, only to animal models. Maybe the information is not available, but this should be stated.
Response 1:Unfortunately, most of the information in this field of research was obtained from animal models and could not always be adequately extrapolated to humans. This indication has been added to the manuscript to the discussion of effects of DNA methylation on the incidence of aneuploidy (lines 588-590 in revised version). Additional data on human has been added to the section about waves of DNA methylation reprogramming (lines 190-191 in revised version).
Point 2:I think that if authors added some figures explaining the methylation processes the manuscript will be easier to understand and more dynamic (for example in point 3).
Response 2: Figure 1 has been added to the manuscript with the mechanisms and key players of DNA methylation and demethylation (lines 119-120 in revised version).
Point 3: In figure 1, in the level of genome methylation graph, I am sure is obvious but I do not understand the birth stages, could you explain this to me, please?
Response 3: The figure has been supplemented with the necessary information namely the left part of the figure shows the level of DNA methylation and the occurrence of aneuploidy in the germ line, and the right part - in somatic cells. In the revised version of manuscript the numbering of figures has been changed. Currently, this is Figure 2.
Point 4: Point 8 is confusing to me, I am not sure if authors are talking about aneuploidies in somatic or embryonic cells.
Response 4: The authors apologize for the incorrect numbering of chapters in the manuscript, which caused misunderstanding. Chapter 8 was intended as a unifying one for subsequent chapters 9-11.
In the revised version of the manuscript, Chapter 8 has been significantly revised to include chapters 9-11, 15-19. As a result, section 6 has been created, which discusses all the issues of the possible relationships of DNA methylation and aneuploidy, summarized in Figure 3.
Point 5: In summary: I am aware that this topic is very complicated and hard to explain. Here relays the difficulty of writing a good and understandable review. Authors have worked very hard and they have recollected a huge amount of information. Nevertheless, in my opinion the order of the information could be improved, to facilitate the reading and understanding. For example, the review should start with the general aspects of methylation (processes, mutation and so on) and then link this information with the generation of aneuploidies in gametes and embryos. It is very interesting to know about the carcinogenesis and animal models, but I think the main point is not clear once you read all the manuscript. I consider that rewriting is necessary to obtain a fluent explanation of the generation of aneuploidies due to methylation issues. In the end, the reader should have to be able to follow the manuscript like a story, not like paragraphs of information that are appearing, and you do not know exactly how to associate them with the rest of the information.
Response 5: In accordance with the recommendations of the reviewer, the manuscript has been significantly revised with the change in the order of presentation of information. The revised manuscript begins with the presentation of the relevance of the problem of aneuploidy in ontogenesis, then discusses the mechanisms of DNA methylation and demethylation, and then traces the mirrored character of the level of DNA methylation and the frequency of aneuploidy kinetics in ontogenesis. Finally, possible mechanisms of aneuploidy influence on DNA methylation and, conversely, the effect of DNA methylation on the occurrence of aneuploidy are considered.
These changes required a significant rearrangement of the chapters with their consolidation. Specifically, Chapters 4, 6, and 12 on gametogenesis have been combined into section 4. Chapters 7, 13, and 14, which dealt with the processes occurring during the cleavage stage, have been combined into section 5. Chapters 9-11 and 15-19 on the mechanisms of the relationship between DNA methylation and aneuploidy have been combined into section 6.
In addition, introductory and final conclusions have been added to individual sections of the manuscript, which the authors hope can make the presentation clearer: lines 142-148, 175-191, 246-250, 377-393, 415-416, 436-437, 457-459, 462-472, 586-590, 612-615.
Minor comments:
Point 6: Line 65: AMA starts 35-38 y-old not 33, and with this the decreasing in the fertility curve.
Point 7: Line 82: The percentages of mosaic keryotypes in humans seems to high. Could authors make sure this data is correct? And, what about mosaicism in preimplantation embryos?
Response 7: These data relate specifically to preimplantation embryos. The sentence is paraphrased to avoid ambiguity. Lines 82-83 in revised version, “humans” has been changed to “human preimplantation embryos”
Point 8: Line 116: reference is missing.
Response 8: Line 245 in revised version, reference has been added.
Point 9: 470: Drosophila should be in italics, and better to write the complete name of the specie D. melanogaster. Does the term yeast only refer to S. cerevisiae or to yeast in general? because if this is the case, it should be stated here.
Response 9: Line 207 in revised version, “Drosophila” and “yeast” have been replaced by “D. melanogaster” and “S. cerevisiae”, respectively
Point 10: DNA methylation crossing over: in order to make this
Response 10: This remark seems to be underwritten
Point 11: Line 508: reference? This part is so short that maybe could be added to other point of the manuscript.
Response 11: The authors apologize for the incorrect numbering of chapters in the manuscript, which caused misunderstanding. Chapter 15 was intended as a unifying one for subsequent chapters 16-19. In the revised version of the manuscript, chapters 15-19 have been merged into one section 6.1.
